# Prebiotic Effect of Polysaccharides and Flavonoids from *Passiflora foetida* Fruits on the Human Intestinal Microbiota Associated with Obesity

**DOI:** 10.3390/foods14183222

**Published:** 2025-09-17

**Authors:** Ya Song, Minqian Zhu, M. Carmen Martínez-Cuesta, Teresa Requena

**Affiliations:** 1Moutai Institute, School of Food Engineering, Renhuai 564507, China; songya_1990@163.com; 2Bioresources Processing Institute of Australia (BioPRIA), Department of Chemical and Biological Engineering, Monash University, Clayton, VIC 3800, Australia; minqian.zhu@monash.edu; 3BFBL Group, Instituto de Investigación en Ciencias de la Alimentación CIAL (CSIC), Campus UAM Cantoblanco, 28049 Madrid, Spain; carmen.martinez@csic.es

**Keywords:** *Passiflora foetida*, polysaccharides, flavonoids, gut microbiota, obesity

## Abstract

*Passiflora foetida* fruit is rich in beneficial polysaccharides and flavonoids. Recent studies have found that these polysaccharides and flavonoids may influence health through interaction with the gut microbiota, for example by modification of the microbial composition or by conversion of the polysaccharides and flavonoids to further bioactive compounds. In the current study, a three-stage dynamic simulator of the human gut microbiota, namely BFBL gut model, inoculated with either normal-weight or obese fecal bacteria, was tested with two aqueous-extracted fractions from *P. foetida* fruit, containing complex dietary *P. foetida* polysaccharides (PFP) and *P. foetida* flavonoids (PFF) mixtures, respectively. Within the context of the gut model, the effects of these interventions on targeted microbial composition as well as metabolite levels were assessed. The results showed that the consumption of PFP and PFF could modulate the microbiota associated to obesity, through regulating the abundance of several microbial groups (*Alistipes*, *Bacteroides*, *Faecalibacterium* and sulphate-reducing bacteria) and enriching the short-chain fatty acids (SCFAs) production, to the levels closer to those in the normal-weight microbiota. Furthermore, this study demonstrated that these complex polysaccharides and flavonoids in the context of an in vitro dynamic gut model showed prebiotic effects on the human intestinal microbiota by modulating some anti-obesity-related bacteria such as *Akkermansia* and *Faecalibacterium* as well as an increase of propionic acid production by the obese microbiota. These microbiota members represent novel targets of *P. foetida* fruit polysaccharides and flavonoids degrading or resistant microbes to be validated under physiological conditions in vivo and further investigated for *P. foetida* fruit beneficial effects.

## 1. Introduction

In recent decades, with increasing evidence demonstrating that human gut microbiota disorders highly associate with a wide range of chronic diseases, the link between human health and the composition and resilience of the gut microbial community has stimulated great attention of scientists [1]. During the co-evolution with the host, the gut microbes have developed unique abilities to digest complex dietary polysaccharides and phytochemicals (polyphenols, saponins, etc.) and convert them into bioactive molecules such as SCFAs and phenolic metabolites [2].

Obesity is a fundamental risk factor for multiple metabolic complications and has become a serious global health issue [3]. Due to the complex pathogenesis, few efficient pharmacotherapies are available in the market for preventing and/or treating obesity. Also, emerging experimental data are revealing that gut microbiota is a potential therapeutic target associated with obesity and related metabolic diseases. Dietary intervention strategies to ameliorate physiological and pathological process of obesity through multiple mechanisms have attracted scientific attention [1]. This strategy has been further supported by some recent investigations of the anti-obesity effects of jamun [4], camu-camu [5], *Rosa roxburghii* Tratt [6] extracts, among others, through modulating gut microbiota community diversity.

*Passiflora foetida* fruit (Passifloraceae) is an edible berry widely consumed in tropical regions, such as Brazil and South China, for treating acute edema, traumatic cornea, conjunctivitis and relieving asthma [7]. Our previous studies suggest that *P. foetida* fruit is rich in polysaccharides and flavonoids and possess various bioactivities in vitro, including antioxidant [7], anti-inflammatory [8] and immunomodulatory [9] effects. Considering the fact that polysaccharides and flavonoids have poor bioavailability in the upper intestinal tract and could reach the large intestine in sufficient amounts to be metabolized by gut microbiota, thereby influencing gut microbiota composition, we hypothesized that extracts of *P. foetida* fruit polysaccharides and flavonoids could influence the composition and/or metabolism of obesity-associated microbiota.

To verify this viewpoint, the current study was designed to investigate whether supplementation with two aqueous-extracted fractions from *P. foetida* fruit could reshape the gut microbial community structure and impact SCFAs production and proportion in a dynamic gastrointestinal simulator (BFBL Gut Model). The potential links between gut microbiota modulation and metabolic benefits of *P. foetida* fruit intake are also discussed herein, which may help to improve scientific understanding of the potential mechanisms of *P. foetida* fruit polysaccharydes and phytochemicals in controlling metabolic diseases.

## 2. Materials and Methods

*P. foetida* fruits (Sanya, Hainan, China) were washed with tap water and distilled water, then dried at 45 °C in an oven. Remove the peels from the dried fruit samples, retain the edible parts, then grind them and sieve them (through a 40-mesh sieve). Pancreatin, DNase and RNase free water, Nessler’s reagent and SCFAs standards (acetic, propionic, and butyric acids) were purchased from Sigma-Aldrich (St. Louis, MO, USA).

### 2.1. Analyses of the Two Aqueous-Extracted Fractions

The acquisition of the two aqueous—soluble fractions was based on our previous research [8,9] with minor modification. To obtain the crude aqueous extracts of *P. foetida* (AEP), the fruits powder (2 kg) was extracted three times with distilled water (powder: solvent, 1:10, *w*/*v*) at 65 °C for 2 h and then centrifuged at 12,000 rpm for 10 min. The supernatants were combined and concentrated in a rotary concentrator under reduced pressure at 50 °C. Then the concentrated AEP was precipitated by adding 10 times the volume of ethanol (96%) overnight at 4 °C. Thus, the *P. foetida* polysaccharide-rich fraction (precipitate, PFP) and flavonoid-rich fraction (supernatant, PFF) were obtained. The PFP was purified by macroporous resin chromatography (AB-8) to remove impurities such as flavonoids, alkaloids, pigments and saponins from the extract, and small molecule impurities were removed by dialysis (3 kDa) [9]. Soluble impurities such as soluble sugars and inorganic salts in the PFF were removed by macroporous resin chromatography (AB-8), and polyphenols and flavonoids were enriched by polyamide chromatography. The chemical composition of these PFP and PFF fractions was analysed by phenol-sulphuric acid assay (total carbohydrate content) [7], Kjeldahl method (total protein content) [10], Folin–Ciocalteu assay (total polyphenol content, TPC), aluminum chloride colorimetry assay (total flavonoid content, TFC) [7], the Association of Official Analytical Chemists method (ash content) [11] and sulfuric acidcarbazole method (uronic acid content) [12], The monosaccharide composition and molecular weight analysis of PFP was performed by the method described in our previous study [9].

For LC/MS analysis, a LTQ Orbitrap mass spectrometer equipped with an electrospray ionization (ESI) source was coupled to a Thermo Dionex Ultimate 3000 UHPLC system (Thermo Fisher Scientific, Waltham, MA, USA). UHPLC separations were executed on a Waters ACQUITY UPLC BEH C18 (2.1 mm × 100 mm, 1.7 μm) column thermo-stated at 40 °C. The mobile phases consisted of water with 0.1% acetic acid (A) and acetonitrile with 0.1% acetic acid (B), and the gradient program was conducted as follows: 0–3 min (5–15% B), 3–11 min (15–30% B), 11–15 min (30–50% B), 15–21 min (50–90% B) and 21–22 min (90–5% B). The sample flow rate was 0.3 mL/min and the injection volume was 2 μL. The MS conditions were as follows: the capillary temperature was 350 °C, the source voltage was set at 3.5 kV, and the sheath gas (N_2_) flow was 35 psi. The Orbitrap mass analyzer was operated in both positive and negative ionization modes. The full MS scans were acquired in the *m*/*z* range of 50–2000 and the MS/MS experiments were set as data-dependent scans. The data were recorded and processed using Xcalibur, Orbitrap Traditional Chinese Medicine Library (OTCML) and Mass Frontier 6.0 software packages (Thermo Fisher Scientific, Waltham, MA, USA).

### 2.2. Dynamic Simulator of the Gastrointestinal Tract

The dynamic gastrointestinal simulator (BFBL Gut Model) was used in the operating mode to work with the units simulating the small intestine (SI) and the ascending (R1), transverse (R2) and descending colon (R3) regions according to the previously published procedures with some modifications [13]. The four vessels were continuously stirred (150 rpm) with an Agimatic N magnetic stirrer with heating (Selecta, Barcelona, Spain) and the temperature was kept at 37 °C. The system was maintained anaerobically through flushing N_2_, and the pH of the colonic reactors was automatically adjusted with the addition of 0.5 M NaOH or 0.5 M HCl to keep values of 5.7 ± 0.2 in the R1, 6.3 ± 0.2 in the R2 and 6.8 ± 0.2 in the R3. These three compartments were filled and pre-conditioned with the nutritive medium that contained arabinogalactan (1 g/L), pectin (2 g/L), xylan (1 g/L), potato starch (3 g/L), glucose (0.4 g/L), yeast extract (3 g/L), peptone (1 g/L), mucin (4 g/L) and L-cysteine (0.5 g/L) in a volume of 250, 400 and 300 mL, respectively. The digestion in the small intestine was simulated in reactor SI by adding 75 mL of nutritive medium (pH 2) mixed with 40 mL artificial pancreatic and bile juice that contained 6 g/L oxgall, 1.9 g/L pancreatin from porcine pancreas and 12.5 g/L NaHCO_3_ to reach the pH 6.5–7.5.

### 2.3. Experimental Protocol Using the BFBL Gut Model

To evaluate the prebiotic effect of PFP and PFF on the intestinal microbiota using the BFBL system, the R1, R2 and R3 compartments were inoculated with 20 mL faeces slurry (20%, *w*/*v*) from five normal weight (BMI between 18.5 and 24.9 kg/m^2^) and five obese (BMI ≥ 30 kg/m^2^) volunteers, respectively, as described previously [13]. Donor gender and BMI distribution and exclusion criteria were previously described. The faecal pooled samples (around 8 g from each donor) were placed into a Stomacher filter bag (filter size 0.28 mm), suspended in 200 mL sodium phosphate buffer 125 (0.1 M, pH 7.0), containing 1 g/L sodium thioglycolate as reducing agent, and homogenized in a Stomacher for 2 min at regular speed.

The development and stabilization of the microbial community until steady-state conditions in the three colon reactors was approached by feeding the small intestine with nutritive medium (75 mL, pH 2) mixed with pancreatic juice (40 mL) three times a day for two weeks. The small intestine digestion was performed during 2 h at 37 °C and the content of the vessel was automatically transferred to the following colonic compartment (R1) at a flow rate of 5 mL/min, which activated the transit of colonic content between the R1, R2 and R3 compartments controlled by level sensors that keep reactor volumes constant. The stabilization of the microbial community until steady-state conditions was evaluated by sampling and measuring the production of SCFAs and ammonium over time. Stability was reached when rates of change of the parameters measured dropped below 10% for each colonic compartment. After the two-week stabilization period (CK) of the colonic microbiota, the BFBL gut model was subjected to an one-week experiment period (PFP) by adding to the nutritive medium 2 g/L PFP, followed by an one-week wash-out period (WO-1), and another one-week test period by adding PFF (2 g/L) to the nutritive medium. After the PFF period, another one-week wash-out period (WO-2) was performed at the end of the experiment (Figure 1). During the whole study, samples were collected daily at regular time points from the three colon vessels and stored at −20 °C until further analysis.

### 2.4. Microbiota Composition Analysis

The real-time quantitative polymerase chain reaction (qPCR) assay was carried out by using SYBR green methodology in a ViiA7 Real-Time PCR System (Life Technologies, Carlsbad, CA, USA). Primers, amplicon size, and amplification conditions for quantification of specific bacterial groups and species were listed in Appendix A. Specifically, the conditions for quantification of *Lactobacillus*, *Bifidobacterium*, *Akkermansia*, *Bacteroides*, *Enterococcus*, *Blautia coccoides-Eubacterium rectale* Cluster XIVa (*B. coccoides*), *Prevotella*, *Enterobacteriaceae*, *Clostridium leptum* subgroup specific cluster IV (*C. leptum*), *Faecalibacterium*, *Ruminococcus* Cluster IV (*Ruminococcus*), *Roseburia*, and *Bilophila* have been described previously [14]. For the analysis of sulphate-reducing bacteria (SRB), *Bilophila*, *Archaea*, *Atopobium*, and *Alistipes* we used the primers and PCR conditions described by Kondo et al. [15], Baldwin et al. [16], Baker et al. [17], Matsuki et al. [18], and Roager et al. [19], respectively. DNA from *Escherichia coli* DH5α, *Lactobacillus plantarum* IFPL935, *Enterococcus faecalis* IFPL382, *Bifidobacterium breve* 29M2, and *Bacteroides fragilis* DSM2151 were used to quantify *Enterobacteriaceae*, *Lactobacillus*, *Enterococcus*, *Bifidobacterium*, and *Bacteroides*, respectively. For all other groups analyzed, samples were quantified using standards derived from targeted cloned genes using the pGEM-T cloning vector system kit (Promega, Madison, WI, USA). Microbial DNA extraction of the samples taken from the R1, R2 and R3 compartments was performed as described by Barroso et al. [14] The DNA yield was measured using a NanoDropH ND-1000 UV spectrophotometer (Thermo Fisher Scientific, Waltham, MA, USA).

### 2.5. Metabolite Production Analysis

The BFBL gut model samples (1 mL) were collected daily from each reactor (R1, R2 and R3) during each period for SCFA and ammonium analyses.

The determination of SCFA was previously described by Barroso et al. [12] Briefly, filtered (0.22 μm) samples were analyzed by a HPLC system equipped with organic acids column (Rezex ROA, 300 × 7.8 mm) (Phenomenex, Macclesfield, UK) and UV-975 detector (Jasco, Tokyo, Japan). SCFAs were separated using a Rezex ROA Organic Acids column (300 × 7.8 mm) (Phenomenex) at 50 °C. The chromatographic conditions for organic acids were 0.005 M sulphuric acid solution, with a flow rate of 0.6 mL/min. The elution profile was monitored at 210 nm and peak identification was carried out by comparison between retention times and standards. The calibration curves of acetic acid, propionic acid and butyric acid in the range concentration of 1 to 100 mM were established.

The content of ammonium was determined using the Nessler’s reagent method with minor modification [20]. Briefly, supernatant (1 μL) from each reactor was diluted by ultrapure water (199 μL) and reacted with Nessler’s reagent (50 μL) for 5 min in the dark. The absorbance was measured at 425 nm using a microplate detector (Varioscan Flash, Thermo Fisher Scientific, Waltham, MA, USA). For preparation of a standard curve, a dilution series of ammonium chloride was prepared in the range of 0 to 20 mM NH_4_^+^. All samples were analyzed in duplicate.

### 2.6. Statistical Analysis

Data of all the analyses are means of three replicates. Significance of all results was evaluated using the statistical software SPSS (version 21.0, SPSS Inc., Chicago, IL, USA) with one-way ANOVA, and individual means were compared using the Tukey’s test (*p* < 0.05).

## 3. Results and Discussion

### 3.1. Chemical Compositions of PFP and PFF

Chemical analysis and monosaccharide composition of PFP are summarized in Appendix A. The results showed that PFP is a high-molecular extract with a molecular weight of 5.25 × 10^4^ Da, and rich in carbohydrates (65.23%), uronic acid (17.05%), flavonoid compounds (12.64%), phenolic compounds (4.27%), and low in ash (1.45%), protein (0.56%). This indicated that there is a certain proportion of bound forms of flavonoids, polyphenols and minerals in the PFP. Moreover, the monosaccharide composition results (Appendix A) demonstrated that PFP was mainly composed of galacturonic acid (28.44%), galactose (26.63%), arabinose (14.87%), mannose (9.30%), glucose (9.26%) and xylose (7.44%). Small amount of fucose and glucuronic acid were also determined in PFP. This composition suggests that PFP is likely a pectic polysaccharide, potentially with a rhamnogalacturonan-I (RG-I) type structure bearing arabinogalactan side chains.

Preliminary quantification of the TPC and TFC showed that another aqueous-extracted fraction PFF contained 40.32 mg gallic equivalents (GAE)/g phenolic and 254.53 mg rutin equivalents (RUE)/g flavonoid. To better understand the composition of PFF, the detection and identification of its components were conducted by UHPLC-ESI–Orbitrap-MS. Comparing with the accurate mass and the fragment ions in the previous reports (Appendix A), the results demonstrated that PFF was a flavonoids-rich fraction. Altogether, a total of 39 constituents, including 8 alkaloids (7.06%), 8 flavones (36.27%), 3 flavonols (0.45%), 13 flavonoid glycosides (44.93%), 3 phenols (5.36%), 2 terpenoids (5.06%), 1 fatty acid (0.32%) and 1 coumarin (0.56%) were identified or tentatively characterized by OTCML. Among them, the highest relative abundance was vicenin II (15.26%), followed by tectorigenin (10.44%) and 4 ′′-O-glucosylvitexin (10.16%), all of which have been confirmed to have potential prebiotic effects [21,22].

### 3.2. The Obesity-Associated Microbiota Stabilized in the BFBL Gut Model

The composition of the intestinal microbial communities and SCFA production during the last three days of the stabilization period in the R1, R2 and R3 compartments were evaluated by quantitative PCR (qPCR) and HPLC, respectively. Figure 2 illustrates the counts of targeted general bacteria and specific phylogenetic and functional groups of normal weight (N) and obesity (O) in different regions of the BFBL system. The results showed that the counts of *Akkermansia*, *Archaea*, *C. leptum*, *Enterobacteriaceae*, *Enterococcus*, *Faecalibacterium*, *Prevotella* and *Ruminococcus* in the obese microbiota were lower (*p* < 0.05) than in the normal-weight microbiota. Meanwhile, obesity-associated microbiota may be represented by *Alistipes*, *Atopobium*, *Bacteroides*, *Lactobacillus* and SRB, as their log copy numbers were found to be significantly increased when compared with the normal-wheight associated microbiota (*p* < 0.05). Overall, our microbiological results agreed with published information describing differences in specific microbial groups associated to obesity [23,24]. These results were highly consistent with the previous studies, which indicated that intestinal *Enterococcus* [25], *Faecalibacterium* [26], *Prevotella* [27], *Ruminococcus* [28] and mainly *Akkermansia* [29,30] abundance correlate inversely with obesity, whereas *Bacteroides* [31] and *Lactobacillus* [32] abundance has been reported to be positively associated to obesity.

### 3.3. The Production of Metabolites During the Stabilization Period of Normal Weight and Obesity Associated Microbiota

The production of main SCFAs (acetic acid, propionic acid and butyric acid), and ammonium by the microbial community during the last three days of the stabilization period in the obese or normal-weight colonic reactors are shown in Figure 3. As expected, both in the obese and normal-weight colonic models, the metabolic activities of the distal colon compartments (R2 and R3) were significantly higher than that of the proximal colon compartment (R1), which may be caused by the accumulation of products in the system and consistent with the results of the previous studies [13,14]. Additionally, compared to normal weight, obesity-associated microbiota showed increases in production of ammonium (1.20, 1.36 and 1.27-fold). Furthermore, obese microbiota resulted in significant decreases in production of acetic acid (1.90, 1.15 and 1.13-fold), propionic acid (3.44, 1.59 and 1.79-fold), and butyric acid (1.31, 1.48 and 1.70-fold) in the R1, R2 and R3 compartments, respectively. The lower production of these SCFAs may be due to the significant depletion of SCFA-producing bacteria, such as *Akkermansia*, *C. leptum*, *Faecalibacterium*, *Prevotella* and *Ruminococcus* (Figure 2). Previous studies revealed that propionic acid is involved in the regulation of appetite and glucose metabolism and had ability to enhance the development of regulatory T cells and to reduce the expansion of proinflammatory Th17 cells [29]. Among the differences in SCFAs between the normal-weight and obese microbiotas, the notable reduction in propionic acid release by the obese microbiota suggest that it might serve as a microbial marker connected to obesity-related metabolic diseases.

### 3.4. Impact of PFP and PFF Feeding on the Obese Microbiota

As displayed in Table 1, the feeding of the BFBL gut model of PFP increased the counts of specific phylogenetic and functional groups both in the obese and normal-weight microbiotas. The 7-day PFP intervention significantly promoted the growth of 9 functional groups, including *Alistipes*, *Atopobium*, *Bacteroides*, *B. coccoides*, *Bilophia*, *C. leptum*, *Enterococcus*, *Faecalibacterium* and SRB, whereas it caused a significant decrease of *Enterobacteriaceae* in the normal-weight microbiota (*p* < 0.05) when compared with the stabilization (CK) period. After the following WO-1 period, 3 functional groups (*Alistipes*, *B. coccoides* and *C. leptum*) remained significantly higher than the CK period (*p* < 0.05). The positive impact of PFP consumption on these beneficial groups, suggest further evaluation of PFP to attenuate colitis [18,33], colorectal cancer [34], or hypercholesterolemic [35] conditions.

Notably, some gut microbial communities associated to obesity were modulated by PFP supplementation to reach levels closer to those in the normal-weight microbiota. Namely, a decrease of *Alistipes* counts and an increase of *Bilophila* and *Faecalibacterium* (*p* < 0.05) were observed during PFP supplementation of the obese microbiota. After the following WO-1 period, the counts of *Bilophila* remained significantly higher than during the CK period (*p* < 0.05), while the decreasing groups (*Alistipes*) returned to the original values in reactor R3. These changes in microbial populations suggest that PFP has potential prebiotic effects to modulate obesity-associated microbiota.

The setup of the experiment included a 7-days feeding of the BFBL gut model with the PFF after the WO-1 period. The PFF test period reduced the upward growth trend of specific phylogenetic and functional groups, both in the obese and normal-weight microbiotas. The intervention of PFF showed significant inhibiting effects on *Enterococcus*, *B. coccoides*, *C. leptum*, *Ruminococcus* and SRB, and remarkable enriching effects on *Lactobacillus* and *Akkermansia* in the normal-weight microbiota model (*p* < 0.05). After the following WO-2 period, the effect of PFF on *Akkermansia* remained, while inhibiting effects of PFF on *Enterococcus*, *B*
*coccoides*, *C leptum*, *Ruminococcus* and SRB were neutralized. *Akkermansia* has been related to metabolic health during dietary polyphenols interventions in obesity, such as the polyphenol-rich cranberry extract and grape polyphenols [28,36,37].

With regarding to the obese microbiota, after the wash-out period (WO-1), PFF supplementation resulted in significant decreases in the counts of *Alistipes*, *Bacteroides* and SRB, while resulted in notable increases in the abundance of *Faecalibacterium* (*p* < 0.05). After the following WO-2 period, these 3 decreasing groups returned to their original values. These results were in accordance with a previous study, which demonstrated that the *Pandanus tectorius* fruit extract (PTF) was effective in ameliorating hyperlipidaemia through selectively enhancing the relative abundance of *Faecalibacterium* and decreasing the relative abundance of *Bacteroides* and *Alistipes* [38]. Thus, the enrichment of *Faecalibacterium* abundance and inhibition of *Bacteroides*, SRB and *Alistipes* abundance demonstrated that PFF could ameliorate the dysbiosis related to obesity.

### 3.5. Impacts of PFP and PFF Consumption on Metabolite Production

The ammonium concentration, a marker for proteolytic activity, was higher in the R2 and R3 reactors than in the R1 during all the experimental setup (Figure 4B–D). Moreover, significant increases in ammonium production (*p* < 0.05) were noticed in the course of the PFP period, when compared with the WO-1 period in all reactors, where the ammonium production showed a marked decrease (*p* < 0.05) to return to the CK values. Among these, the ability of PFP to promote the ammonium production in R1 (increased 1.51-fold) was stronger than that of R2 (increased 1.06-fold) and R3 (increased 1.01-fold) in the normal-weight microbiota. As for the obese microbiota, the ammonium production during the PFP period showed increases of 1.19-fold, 1.19-fold and 1.09-fold, respectively, when compared with the CK period and were also significantly higher than that of the WO-1 period.

During the experimental setup, the 7-day PFF intervention enriched the ammonium production in R1 (1.42 times), R2 (1.23 times) and R3 (1.15 times) in the normal-weight microbiota when compared with the WO-1 period. As for the obese microbiota, the ability of PFF to promote the ammonium production in these colonic compartments (R1: 1.40-fold, R2: 1.32-fold, and R3: 1.53-fold) was stronger than that of PFP (1.19, 1.19 and 1.09-fold, respectively).

Another relevant metabolic activity of the colonic microbiota is the formation of SCFAs. During the PFP feeding period, both in the normal-weight and obese colonic microbiotas, the concentration of total SCFAs (Figure 4A) of the distal colonic compartments (R2 and R3) remained higher than that of the proximal colonic compartment (R1). Regarding the changes in formation of specific SCFAs (Figure 4B–D), the supplementation of PFP to the colonic microbiota had an impact on the concentration of the propionic acid production both in the normal-weight and obese microbiotas in all the reactors. The propionic acid production of normal-weight colonic microbiota in reactors R1 and R2 was significantly higher than that of CK and WO-1 periods (*p* < 0.05). The increases in propionic acid could be associated to the fermentation of galactose and galacturonic acid in PFP and the enrichment of propionic acid-producing bacteria such as *Bacteroides* (Table 1). Meanwhile, increasing the microbiota production of propionate or its delivery in the colon has been considered as an attractive solution to improve metabolic disorders [39].

Similar to PFP, the supplementation of PFF notably increased the propionic acid production in the normal-weight colonic model (*p* < 0.05), but it did not make differences (*p* > 0.05) on acetic acid and butyric acid formation. The result could be associated with the observed enrichment of propionic acid-producing bacteria such as *Akkemansia* (Table 1). Moreover, notable increases (*p* < 0.05) of acetic acid, propionic acid, and butyric acid production were observed when the obese microbiota was supplemented with PFF (Figure 4). These results suggest that the increase in SCFAs was primarily due to the fermentation of the flavonoid glycosides in the PFF.

## 4. Conclusions

This study has for the first time investigated the fermentation behavior of polysaccharide-rich fraction (PFP) and flavonoid-rich fraction (PFF) from aqueous extract of *P. foetida* fruits by normal-weight and obese human microbiotas in vitro through the BFBL dynamic gastrointestinal simulator. In the BFBL model, the feeding with PFP and PFF could restore the initial differences found between the normal-weight and obese microbiotas (*Alistipes*, *Bacteroides*, *Faecalibacterium* and SRB). These *P. foetida* fractions caused as well an increase of propionic acid production by the obese microbiota. Furthermore, the PFP and PFF interventions showed prebiotic effects on the human intestinal microbiota by modulating some anti-obesity-related bacteria such as *Akkermansia* and *Faecalibacterium*. These results suggest that *P. foetida* fruits might potentially be explored as an effective prebiotic for the treatment and/or prevention of obesity and related metabolic diseases. However, compared with in vivo studies, the in vitro simulator used in this study has inherent limitations, especially the lack of immune/epithelial cross-interaction and physiological variables. Therefore, in our subsequent research, in vivo intervention trials need to be adopted to confirm our findings. Furthermore, the dietary intervention plant extracts used in this study are mixtures, and their specific composition and detailed structure still require further analysis in subsequent research.

## Figures and Tables

**Figure 1 foods-14-03222-f001:**
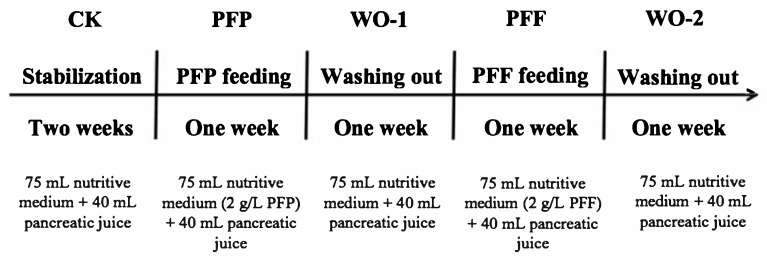
Experimental protocol using the BFBL dynamic gastrointestinal simulator.

**Figure 2 foods-14-03222-f002:**
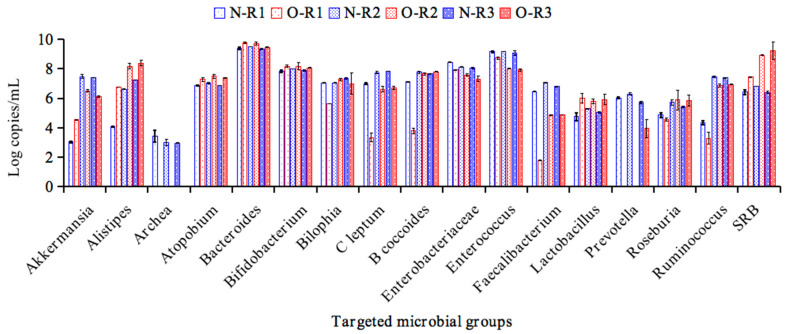
qPCR values (log copy number/mL) of the targeted microbial groups in ascending (R1), transverse (R2) and descending (R3) colon vessels of the BFBL gut model after stabilization (CK stage). N = normal weight; O = obesity.

**Figure 3 foods-14-03222-f003:**
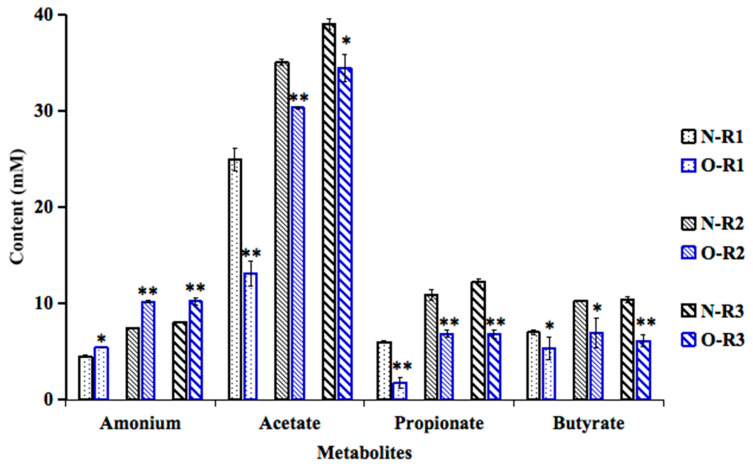
Production of ammonium, acetic acid, propionic acid and butyric acid by the stabilized microbiotas in ascending (R1), transverse (R2) and descending (R3) colon vessels of the BFBL gut model. N = normal weight; O = obesity. * denotes the significance level of the mean value between two comparison groups. * represents *p* ≤ 0.05 and ** represents *p* < 0.01.

**Figure 4 foods-14-03222-f004:**
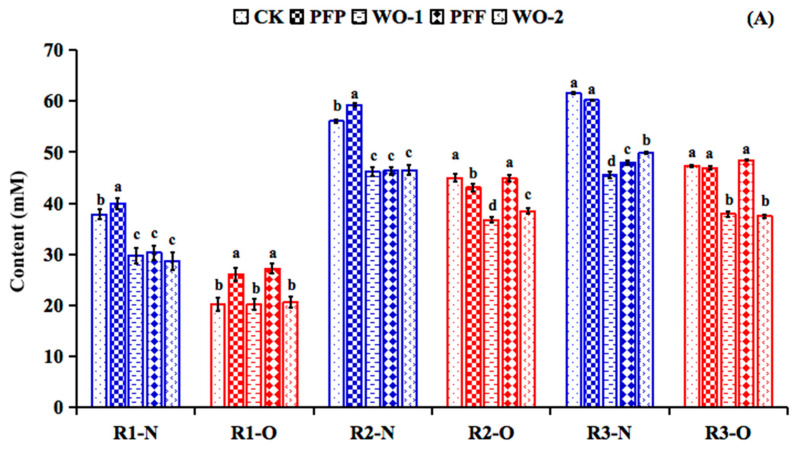
Changes in concentration (mM; mean ± SD) of total SCFA (**A**), acetate, propionate, butyrate and ammonium in the R1 (**B**), R2 (**C**) and R3 (**D**) of the BFBL gut model at the end of the stabilization period (CK), experiment periods (PFP and PFF) and wash-out periods (WO-1 and WO-2). N = normal weight, represented by blue columns; O = obesity, represented by red columns. Different letters indicate significant difference among the values of mean value ± standard error (*p* < 0.05).

**Table 1 foods-14-03222-t001:** Mean ± SD of quantitative PCR counts (log copy number/mL) for the different microbial groups analysed in the ascending (R1), transverse (R2) and descending colon (R3) of the normal weight (N) and obesity (O) microbiotas during the last three days of each experiment period (CK, PFP, WO-1, PFF, WO-2). Data are means of three replicates ± SD. Averages with different letters in the same row represent statistical difference in the Tukey’s test (*p* < 0.05) for the microbial group.

Bacterial Group	Regions	CK	PFP	WO-1	PFF	WO-2
*Akkermansia*	R1-N	3.04 ± 0.05 ^d^	3.09 ± 0.09 ^d^	4.25 ± 0.07 ^c^	4.42 ± 0.06 ^b^	4.70 ± 0.13 ^a^
	R1-O	4.54 ± 0.02 ^a^	3.99 ± 0.09 ^c^	4.33 ± 0.25 ^ab^	4.09 ± 0.04 ^bc^	3.86 ± 0.06 ^c^
	R2-N	7.48 ± 0.13 ^b^	7.77 ± 0.13 ^b^	7.45 ± 0.17 ^b^	8.22 ± 0.13 ^a^	8.43 ± 0.06 ^a^
	R2-O	6.50 ± 0.07 ^bc^	6.05 ± 0.55 ^c^	6.93 ± 0.05 ^b^	7.64 ± 0.20 ^a^	7.77 ± 0.14 ^a^
	R3-N	7.39 ± 0.01 ^c^	7.87 ± 0.19 ^b^	7.73 ± 0.03 ^b^	8.31 ± 0.01 ^a^	8.35 ± 0.02 ^a^
	R3-O	6.12 ± 0.03 ^c^	6.25 ± 0.07 ^c^	6.55 ± 0.14 ^b^	7.62 ± 0.19 ^a^	7.61 ± 0.07 ^a^
*Alistipes*	R1-N	4.07 ± 0.03 ^b^	5.01 ± 0.03 ^a^	3.54 ± 0.05 ^c^	3.91 ± 0.24 ^b^	3.47 ± 0.02 ^c^
	R1-O	6.76 ± 0.01 ^a^	5.85 ± 0.47 ^b^	5.17 ± 0.09 ^b^	5.54 ± 0.08 ^b^	5.56 ± 0.33 ^b^
	R2-N	6.62 ± 0.03 ^c^	7.60 ± 0.21 ^b^	7.96 ± 0.04^a^	7.79 ± 0.08 ^b^	7.92 ± 0.02 ^a^
	R2-O	8.17 ± 0.18 ^a^	7.53 ± 0.09 ^b^	7.77 ± 0.11 ^b^	7.54 ± 0.15 ^b^	7.79 ± 0.13 ^b^
	R3-N	7.23 ± 0.01 ^d^	7.66 ± 0.02 ^b^	7.94 ± 0.02^a^	7.50 ± 0.04 ^c^	7.67 ± 0.11 ^b^
	R3-O	8.39 ± 0.13 ^a^	7.47 ± 0.21 ^b^	8.25 ± 0.20 ^a^	7.43 ± 0.18 ^b^	8.12 ± 0.15 ^a^
*Archaea*	R1-N	3.44 ± 0.39 ^ab^	2.98 ± 0.19 ^bc^	3.64 ± 0.05 ^a^	2.50 ± 0.21 ^c^	3.46 ± 0.24 ^ab^
	R1-O	ND ^c^	2.16 ± 0.58 ^b^	1.94 ± 0.44 ^b^	2.49 ± 0.38 ^b^	3.72 ± 0.18 ^a^
	R2-N	2.99 ± 0.21	3.36 ± 0.14	2.85 ± 0.50	2.86 ± 0.48	3.40 ± 0.24
	R2-O	ND ^d^	1.57 ± 0.09 ^c^	2.26 ± 0.29 ^b^	2.15 ± 0.25 ^b^	3.26 ± 0.41 ^a^
	R3-N	2.97 ± 0.01 ^bc^	3.77 ± 0.35 ^a^	2.92 ± 0.16 ^c^	2.60 ± 0.05 ^c^	3.36 ± 0.08 ^b^
	R3-O	ND ^c^	1.15 ± 0.17 ^b^	1.37 ± 0.62 ^b^	1.85 ± 0.25 ^ab^	2.42 ± 0.35 ^a^
*Atopobium*	R1-N	6.87 ± 0.03 ^d^	7.59 ± 0.13 ^a^	7.36 ± 0.03 ^b^	7.06 ± 0.14 ^c^	7.31 ± 0.02 ^b^
	R1-O	7.28 ± 0.30	7.36 ± 0.07	7.66 ± 0.15	7.53 ± 0.04	7.48 ± 0.07
	R2-N	7.00 ± 0.06	7.60 ± 0.09	7.12 ± 0.04	7.20 ± 0.20	7.18 ± 0.02
	R2-O	7.49 ± 0.21 ^b^	7.60 ± 0.09 ^b^	7.65 ± 0.01 ^a^	7.43 ± 0.02 ^b^	7.49 ± 0.01 ^ab^
	R3-N	6.85 ± 0.01 ^c^	7.30 ± 0.05 ^a^	7.03 ± 0.01 ^b^	7.05 ± 0.02 ^b^	7.09 ± 0.05 ^b^
	R3-O	7.39 ± 0.03 ^b^	7.39 ± 0.03 ^b^	7.80 ± 0.12 ^a^	7.28 ± 0.13 ^b^	7.66 ± 0.29 ^ab^
*B. coccoides*	R1-N	7.12 ± 0.02 ^d^	7.98 ± 0.02 ^a^	7.67 ± 0.13 ^b^	7.42 ± 0.14 ^c^	7.93 ± 0.12 ^a^
	R1-O	3.81 ± 0.17	4.22 ± 0.15	4.03 ± 0.25	3.75 ± 0.24	4.10 ± 0.20
	R2-N	7.76 ± 0.09 ^b^	8.02 ± 0.12 ^a^	7.97 ± 0.02 ^a^	7.79 ± 0.12 ^ab^	7.98 ± 0.01 ^a^
	R2-O	7.65 ± 0.07 ^b^	7.95 ± 0.02 ^a^	7.76 ± 0.12 ^b^	7.61 ± 0.07 ^b^	7.63 ± 0.01 ^b^
	R3-N	7.66 ± 0.02 ^b^	7.99 ± 0.18 ^a^	7.89 ± 0.01 ^a^	7.64 ± 0.07 ^b^	7.86 ± 0.03 ^a^
	R3-O	7.80 ± 0.01 ^a^	7.56 ± 0.07 ^ab^	7.48 ± 0.23 ^b^	7.52 ± 0.07 ^ab^	7.18 ± 0.09 ^c^
*Bacteroides*	R1-N	9.39 ± 0.06 ^b^	9.79 ± 0.10 ^a^	9.81 ± 0.06 ^a^	9.54 ± 0.16 ^b^	9.80 ± 0.03 ^a^
	R1-O	9.77 ± 0.06 ^ab^	9.60 ± 0.04 ^bc^	9.88 ± 0.03 ^a^	9.51 ± 0.15 ^c^	9.70 ± 0.10 ^abc^
	R2-N	9.50 ± 0.01	9.63 ± 0.22	9.68 ± 0.01	9.60 ± 0.19	9.67 ± 0.02
	R2-O	9.71 ± 0.10 ^ab^	9.55 ± 0.10 ^bc^	9.84 ± 0.03 ^a^	9.42 ± 0.10 ^c^	9.71 ± 0.12 ^ab^
	R3-N	9.34 ± 0.03	9.63 ± 0.24	9.56 ± 0.01	9.59 ± 0.11	9.56 ± 0.03
	R3-O	9.47 ± 0.04 ^ab^	9.14 ± 0.04 ^c^	9.60 ± 0.02 ^a^	9.14 ± 0.20 ^c^	9.30 ± 0.05 ^bc^
*Bifidobacterium*	R1-N	7.81 ± 0.08 ^c^	8.31 ± 0.12 ^a^	8.32 ± 0.05 ^a^	7.72 ± 0.05 ^d^	8.09 ± 0.06 ^b^
	R1-O	8.17 ± 0.08 ^ab^	8.20 ± 0.06 ^ab^	8.34 ± 0.18 ^a^	8.22 ± 0.04 ^ab^	8.01 ± 0.25 ^b^
	R2-N	7.99 ± 0.01	8.11 ± 0.15	8.21 ± 0.01	7.84 ± 0.61	8.07 ± 0.06
	R2-O	8.18 ± 0.26	8.37 ± 0.20	8.24 ± 0.24	8.10 ± 0.14	8.04 ± 0.01
	R3-N	7.88 ± 0.03 ^b^	8.23 ± 0.23 ^a^	8.00 ± 0.15 ^ab^	8.12 ± 0.07 ^ab^	8.02 ± 0.05 ^ab^
	R3-O	8.06 ± 0.03	8.04 ± 0.07	8.08 ± 0.35	7.87 ± 0.18	7.83 ± 0.18
*Bilophila*	R1-N	7.05 ± 0.01 ^b^	7.47 ± 0.03 ^a^	7.42 ± 0.04 ^a^	7.16 ± 0.24 ^b^	7.56 ± 0.03 ^a^
	R1-O	5.62 ± 0.01 ^d^	7.10 ± 0.16 ^b^	7.56 ± 0.02 ^a^	7.17 ± 0.10 ^b^	6.35 ± 0.04 ^c^
	R2-N	7.04 ± 0.02 ^b^	7.46 ± 0.11 ^a^	7.36 ± 0.01 ^a^	7.35 ± 0.03 ^a^	7.45 ± 0.01 ^a^
	R2-O	7.29 ± 0.05 ^b^	7.45 ± 0.06 ^a^	7.49 ± 0.13 ^a^	7.39 ± 0.08 ^a^	6.66 ± 0.23 ^c^
	R3-N	7.35 ± 0.05	7.45 ± 0.23	7.36 ± 0.06	7.34 ± 0.10	7.43 ± 0.03
	R3-O	6.99 ± 0.73 ^c^	7.39 ± 0.01 ^ab^	7.65 ± 0.04 ^a^	7.28 ± 0.26 ^bc^	6.28 ± 0.01 ^d^
*C. leptum*	R1-N	7.00 ± 0.06 ^bc^	7.48 ± 0.09 ^a^	7.35 ± 0.01 ^ab^	6.66 ± 0.44 ^c^	7.26 ± 0.11 ^ab^
	R1-O	3.34 ± 0.22	3.11 ± 0.30	3.55 ± 0.28	3.15 ± 0.19	3.10 ± 0.18
	R2-N	7.74 ± 0.08 ^ab^	8.02 ± 0.20 ^a^	7.99 ± 0.01 ^a^	7.79 ± 0.17 ^ab^	7.60 ± 0.04 ^b^
	R2-O	6.61 ± 0.17	6.84 ± 0.47	6.82 ± 0.10	6.59 ± 0.09	6.69 ± 0.02
	R3-N	7.83 ± 0.01 ^bc^	8.12 ± 0.12 ^a^	7.94 ± 0.06 ^b^	7.77 ± 0.08 ^c^	7.55 ± 0.05 ^d^
	R3-O	6.71 ± 0.12	6.41 ± 0.37	6.82 ± 0.30	6.60 ± 0.28	6.77 ± 0.13
*Enterobacteriaceae*	R1-N	8.44 ± 0.01 ^a^	8.03 ± 0.12 ^bc^	8.31 ± 0.11 ^ab^	7.79 ± 0.33 ^c^	8.23 ± 0.05 ^ab^
	R1-O	7.92 ± 0.02 ^a^	7.36 ± 0.13 ^c^	7.41 ± 0.15 ^c^	7.40 ± 0.09 ^c^	7.66 ± 0.02 ^b^
	R2-N	8.12 ± 0.03 ^a^	7.78 ± 0.09 ^b^	7.97 ± 0.06 ^ab^	7.87 ± 0.25 ^b^	7.85 ± 0.04 ^b^
	R2-O	7.59 ± 0.10 ^a^	7.01 ± 0.23 ^b^	7.36 ± 0.05 ^a^	7.36 ± 0.09 ^a^	7.45 ± 0.06 ^a^
	R3-N	8.06 ± 0.05 ^a^	7.93 ± 0.07 ^b^	7.86 ± 0.04 ^b^	7.87 ± 0.02 ^b^	7.92 ± 0.01 ^b^
	R3-O	7.32 ± 0.17 ^a^	6.82 ± 0.09 ^b^	7.29 ± 0.02 ^a^	7.22 ± 0.15 ^a^	7.28 ± 0.15 ^a^
*Enterococcus*	R1-N	9.16 ± 0.04 ^d^	10.00 ± 0.40 ^b^	9.66 ± 0.04 ^c^	9.32 ± 0.18 ^d^	11.63 ± 0.06 ^a^
	R1-O	8.73 ± 0.06 ^a^	6.04 ± 0.03 ^d^	8.65 ± 0.05 ^a^	8.45 ± 0.02 ^b^	7.91 ± 0.15 ^c^
	R2-N	9.17 ± 0.01 ^cd^	9.87 ± 0.10 ^b^	9.24 ± 0.26 ^bc^	8.88 ± 0.51 ^d^	10.39 ± 0.01 ^a^
	R2-O	8.02 ± 0.02 ^bc^	6.36 ± 0.10 ^d^	8.60 ± 0.14 ^a^	8.20 ± 0.07 ^b^	7.80 ± 0.30 ^c^
	R3-N	9.06 ± 0.15 ^bc^	10.03 ± 0.13 ^a^	9.40 ± 0.01 ^b^	8.89 ± 0.39 ^c^	9.87 ± 0.05 ^a^
	R3-O	7.93 ± 0.08 ^b^	6.93 ± 0.19 ^d^	8.44 ± 0.04 ^a^	7.83 ± 0.01 ^b^	7.47 ± 0.05 ^c^
*Faecalibacterium*	R1-N	6.46 ± 0.02 ^b^	6.97 ± 0.12 ^a^	6.53 ± 0.05 ^b^	6.22 ± 0.13 ^c^	6.14 ± 0.02 ^c^
	R1-O	1.78 ± 0.02 ^c^	2.78 ± 0.14 ^b^	1.87 ± 0.07 ^c^	3.14 ± 0.11 ^a^	3.33 ± 0.15 ^a^
	R2-N	7.06 ± 0.01 ^a^	7.00 ± 0.24 ^ab^	6.77 ± 0.03 ^b^	6.55 ± 0.27 ^c^	6.63 ± 0.01 ^c^
	R2-O	4.86 ± 0.04 ^c^	5.72 ± 0.14 ^a^	5.25 ± 0.10 ^b^	5.60 ± 0.09 ^a^	4.94 ± 0.08 ^c^
	R3-N	6.79 ± 0.01 ^b^	6.99 ± 0.10 ^a^	6.37 ± 0.04 ^d^	6.66 ± 0.04 ^c^	6.44 ± 0.01 ^d^
	R3-O	4.87 ± 0.01 ^b^	5.25 ± 0.06 ^a^	5.01 ± 0.11 ^b^	5.21 ± 0.06 ^a^	4.96 ± 0.11 ^ab^
*Lactobacillus*	R1-N	4.75 ± 0.25 ^c^	4.68 ± 0.12 ^c^	4.77 ± 0.08 ^c^	5.09 ± 0.11 ^b^	5.37 ± 0.05 ^a^
	R1-O	6.00 ± 0.30 ^b^	5.01 ± 0.21 ^c^	7.31 ± 0.21 ^a^	7.25 ± 0.20 ^a^	7.24 ± 0.03 ^a^
	R2-N	5.30 ± 0.03 ^a^	4.68 ± 0.23 ^b^	4.95 ± 0.10 ^b^	5.46 ± 0.08 ^a^	4.91 ± 0.06 ^b^
	R2-O	5.81 ± 0.47 ^b^	5.33 ± 0.15 ^b^	7.11 ± 0.23 ^a^	7.07 ± 0.29 ^a^	7.02 ± 0.26 ^a^
	R3-N	5.04 ± 0.03 ^c^	4.55 ± 0.07 ^e^	4.72 ± 0.02 ^d^	5.28 ± 0.17 ^b^	5.82 ± 0.01 ^a^
	R3-O	5.92 ± 0.35 ^b^	4.82 ± 0.31 ^c^	6.94 ± 0.50 ^a^	7.05 ± 0.29 ^a^	6.99 ± 0.08 ^a^
*Prevotella*	R1-N	6.02 ± 0.06 ^a^	5.60 ± 0.06 ^b^	5.52 ± 0.06 ^b^	5.17 ± 0.18 ^c^	5.55 ± 0.10 ^b^
	R1-O	ND ^b^	4.33 ± 0.49 ^a^	4.90 ± 0.29 ^a^	4.14 ± 0.57 ^a^	4.34 ± 0.50 ^a^
	R2-N	6.30 ± 0.06 ^a^	5.80 ± 0.21 ^b^	5.41 ± 0.19 ^b^	5.32 ± 0.28 ^b^	5.59 ± 0.13 ^b^
	R2-O	ND ^b^	4.12 ± 0.54 ^a^	4.54 ± 0.23 ^a^	4.54 ± 0.15 ^a^	4.29 ± 0.09 ^a^
	R3-N	5.71 ± 0.07 ^ab^	5.79 ± 0.01 ^a^	5.19 ± 0.11 ^d^	5.37 ± 0.16 ^cd^	5.53 ± 0.12 ^bc^
	R3-O	3.95 ± 0.43	4.65 ± 0.54	4.33 ± 0.13	4.40 ± 0.26	4.09 ± 0.15
*Roseburia*	R1-N	4.86 ± 0.15	5.42 ± 0.15	5.57 ± 0.01	5.19 ± 0.71	5.58 ± 0.05
	R1-O	4.56 ± 0.10 ^a^	3.75 ± 0.29 ^b^	4.32 ± 0.19 ^a^	3.78 ± 0.05 ^b^	3.72 ± 0.18 ^b^
	R2-N	5.71 ± 0.17	5.75 ± 0.16	5.80 ± 0.01	5.77 ± 0.81	6.24 ± 0.01
	R2-O	5.88 ± 0.66	5.39 ± 0.32	5.76 ± 0.17	5.56 ± 0.21	5.73 ± 0.26
	R3-N	5.41 ± 0.06	5.73 ± 0.16	5.62 ± 0.03	5.70 ± 0.49	5.97 ± 0.07
	R3-O	5.83 ± 0.37	5.13 ± 0.17	5.91 ± 0.27	5.32 ± 0.19	5.63 ± 0.44
*Ruminococcus*	R1-N	4.35 ± 0.13 ^ab^	4.15 ± 0.12 ^b^	4.58 ± 0.21 ^a^	3.75 ± 0.02 ^c^	4.26 ± 0.06 ^b^
	R1-O	3.29 ± 0.40	3.62 ± 0.25	3.73 ± 0.21	3.58 ± 0.34	3.72 ± 0.03
	R2-N	7.45 ± 0.06 ^ab^	7.58 ± 0.20 ^a^	7.53 ± 0.03 ^a^	7.00 ± 0.22 ^c^	7.05 ± 0.19 ^bc^
	R2-O	6.87 ± 0.12 ^a^	6.92 ± 0.07 ^a^	6.78 ± 0.18 ^a^	6.41 ± 0.04 ^b^	6.40 ± 0.03 ^b^
	R3-N	7.39 ± 0.03 ^b^	7.72 ± 0.14 ^a^	7.52 ± 0.03 ^ab^	7.06 ± 0.10 ^c^	6.95 ± 0.09 ^c^
	R3-O	6.94 ± 0.02 ^a^	7.09 ± 0.07 ^a^	6.92 ± 0.39 ^a^	6.78 ± 0.20 ^ab^	6.37 ± 0.06 ^b^
SRB	R1-N	6.42 ± 0.07 ^b^	7.95 ± 0.06 ^a^	8.79 ± 0.19 ^a^	8.27 ± 0.62 ^a^	9.17 ± 0.02 ^a^
	R1-O	7.43 ± 0.02 ^e^	8.17 ± 0.02 ^d^	9.08 ± 0.01 ^a^	8.61 ± 0.02 ^c^	8.94 ± 0.10 ^b^
	R2-N	6.83 ± 0.01 ^c^	7.88 ± 0.78 ^b^	9.20 ± 0.13 ^a^	8.88 ± 0.53 ^a^	9.08 ± 0.04 ^a^
	R2-O	8.93 ± 0.04 ^ab^	9.09 ± 0.04 ^a^	9.08 ± 0.05 ^a^	8.77 ± 0.15 ^b^	8.86 ± 0.09 ^b^
	R3-N	6.41 ± 0.10 ^d^	9.04 ± 0.66 ^bc^	9.17 ± 0.05 ^b^	8.41 ± 0.51 ^c^	9.43 ± 0.11 ^a^
	R3-O	9.24 ± 0.60	9.00 ± 0.06	9.14 ± 0.04	8.56 ± 0.15	9.52 ± 0.71

ND, not detected. Different letters in same line show significant difference among the values of mean value ± standard error (*p* < 0.05). SRB, sulphate-reducing bacteria.

## Data Availability

The original contributions presented in the study are included in the article/Appendix A, further inquiries can be directed to the corresponding author/s.

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
