# Peer review of "Prebiotic Effect of Polysaccharides and Flavonoids from Passiflora foetida Fruits on the Human Intestinal Microbiota Associated with Obesity"

_foods, 2025, doi:10.3390/foods14183222_

Round 1
Reviewer 1 Report
Comments and Suggestions for Authors
- The Introduction is well written and clearly presents the rationale for this study, supported by recent literature.
-In the “Materials and Methods” section, it is unclear which part of the plant was used (peel, seed, pulp, outer layer, or edible portion). Please clarify, including these details.
- The methods are thoroughly described, and references are cited appropriately.
- In the supplementary material, I suggest formatting the document in “landscape” orientation, as it is not possible to properly view all the table data in its current layout.
- In Figure 2, it is difficult to visually differentiate the treatments. I recommend adjusting the bar colors to better match the corresponding legend.
Author Response
Comments 1: The Introduction is well written and clearly presents the rationale for this study, supported by recent literature.
Response 1: Thank you for your positive comment on the introduction of our manuscript. We are pleased that you found it to be well-written and clear.
Comments 2: In the “Materials and Methods” section, it is unclear which part of the plant was used (peel, seed, pulp, outer layer, or edible portion). Please clarify, including these details.
Response 2: We thank the reviewer for this important comment. In our experiments, we used the edible portion of the fruit. Specifically, the peel was removed, and only the inner edible pulp and seeds were retained for analysis. We have now revised the “Materials and Methods” section to explicitly state: " Remove the peels from the dried fruit samples, retain the edible parts, then grind them and sieve them (through a 40-mesh sieve)."
Comments 3: The methods are thoroughly described, and references are cited appropriately.
Response 3: We thank the reviewer for their kind comment regarding the Methods section.
Comments 4: In the supplementary material, I suggest formatting the document in “landscape” orientation, as it is not possible to properly view all the table data in its current layout.
Response 4: We thank the reviewer for this excellent suggestion. We agree that the landscape orientation significantly improves the readability of the supplementary table. We have reformatted the supplementary material accordingly and have submitted the updated file.
Comments 5: In Figure 2, it is difficult to visually differentiate the treatments. I recommend adjusting the bar colors to better match the corresponding legend.
Response 5: We appreciate the reviewer's feedback on the clarity of Figure 2. The bar colors have been modified to provide better visual contrast and to align precisely with the legend.
Reviewer 2 Report
Comments and Suggestions for Authors
The paper looks interesting by means of main goals as well as a partial design of the experiment. Nowadays the main goal of natural and food related sciences is to explore the world of plants, fungus etc. in order to established the sources a preparation for common diseases. This is the second branch of research on human wellness and health besides the synthetic route for medicine treatment. Based on this the aim of the research is clear and is the continuation of earlier research. In my opinion however the research has some basic drawbacks especially at the first stages of experiment. Those may be divided in two main groups:
- Authors decided to experiment on one extract only. It is 2 hours of extraction at 65oC. Why such condition has been chosen? Does Authors did any optimisation experiment (using e.g. RSM). So explanation why 65oC and why 2 hours. Additionally the methodology of extraction is not clear sometimes: ” the fruits powder was extracted with hot water at 65 °C for 2 h and concentrated”. Concentrated how and to what concentration?
- Second part deals with polysaccharide fraction. As far as I understand “the polymer” is highly ionic by means of structure. The questions are. Is this a polymer? What is a degree of polymerisation? What is a molecular mass? And also if the chain contains plenty (8) different units what is the structure of such copolymer. Bloc, comb, other structure? I believe there are the crucial question that needs to be answered to understand the phenomena observed especially when we think about influence on human body. Additionally the fraction contains 55% of carbohydrates (probably mono, oligo and maybe polysaccharides) and 0,4% of protein. What kind of compounds are the rest 44%? It is also crucial.
It needs deep upgrade so major revision is my decision on presented paper
Author Response
Comments 1: Authors decided to experiment on one extract only. It is 2 hours of extraction at 65°C. Why such condition has been chosen? Does Authors did any optimisation experiment (using e.g. RSM). So explanation why 65°C and why 2 hours. Additionally the methodology of extraction is not clear sometimes: ” the fruits powder was extracted with hot water at 65 °C for 2 h and concentrated”. Concentrated how and to what concentration?
Response 1: We thank the reviewer for raising these important points. The conditions (65°C for 2 hours) were not chosen arbitrarily but were directly adopted from our previously published and optimized protocols. As now more clearly stated in the revised manuscript, this exact methodology is described in detail in References 8 and 9, which are our prior studies focused specifically on the optimization of extraction, separation, and purification processes for these exact aqueous components. The extract was concentrated using a rotary evaporator under reduced pressure at 50°C to one-tenth of its original volume. We have clarified this in the revised manuscript.
Comments 2: Second part deals with polysaccharide fraction. As far as I understand “the polymer” is highly ionic by means of structure. The questions are. Is this a polymer? What is a degree of polymerisation? What is a molecular mass? And also if the chain contains plenty (8) different units what is the structure of such copolymer. Bloc, comb, other structure? I believe there are the crucial question that needs to be answered to understand the phenomena observed especially when we think about influence on human body. Additionally the fraction contains 55% of carbohydrates (probably mono, oligo and maybe polysaccharides) and 0,4% of protein. What kind of compounds are the rest 44%? It is also crucial.
Response 2: We are grateful to this reviewer for raising these insightful and crucial questions. Firstly, we confirm that this is a polysaccharide polymer. During the preparation of PFP, we used a dialysis bag with a cut-off of 3 kDa for dialysis to ensure its large molecular nature. Regarding its high ionicity, it is mainly due to the high content (over 30%) of glucuronic acid. Finally, we speculate that the remaining approximately 44% is mainly composed of bound phenolic compounds, pigments, and minerals. This is a typical characteristic of the polysaccharide components extracted in this way. We have added this explanation in the "Results" section.
Reviewer 3 Report
Comments and Suggestions for Authors
First and foremost, I want to congratulate you on a well-executed and important study. You present the first in vitro evaluation of Passiflora foetida fruit polysaccharide-rich (PFP) and flavonoid-rich (PFF) fractions on obesity-associated human gut communities using the dynamic BFBL simulator. Your work convincingly shows that both PFP and PFF shift an obesity-linked microbial profile toward one resembling a healthy, normal-weight state, enrich key short-chain fatty acids (notably propionate), and selectively modulate taxa such as Akkermansia, Faecalibacterium, Alistipes, Bacteroides, and sulfate-reducing bacteria. The multidisciplinary approach, combining thorough chemical characterization, quantitative PCR tracking of nearly twenty phylogenetic/functional groups, and metabolic profiling, represents a commendable integration of food chemistry and gut microbiology.
Below I outline a few suggestions to strengthen clarity, reproducibility, and the manuscript’s broader impact. I believe none of these points undermine the validity of your conclusions; addressing them will simply make your valuable findings even more accessible to readers.
Clarify Donor Selection and Pooling- Please provide a brief description of donor demographics (age range, gender distribution) and any exclusion criteria (recent antibiotic use, chronic illness). In the Methods, you describe pooling fecal material from five normal-weight and five obese individuals. Adding a sentence on how you ensured homogeneity in the pool (e.g., same weight per donor, aseptic mixing protocol) will help readers reproduce the inocula preparation.
2. Define Abbreviations at First Use- Although common in this field, please ensure that every abbreviation (e.g., PFP, PFF, SCFAs, BFBL) is defined upon its first appearance in the text.
3. Enhance Description of Chemical Characterization- You report that PFP is enriched in galacturonic acid, galactose, arabinose, and so on, and that PFF contains 39 tentatively identified compounds. It would benefit the reader to:
Include a one- to two-sentence rationale for why macroporous resin and polyamide chromatography were chosen for PFF versus PFP purification.
Highlight in the main text (not only the Supplement) the three or four major flavonoid glycosides that likely drive the prebiotic effects, including their approximate relative abundance.
4. Discuss Model Limitations and In Vivo Translation- Your Discussion touches on the promise of P. foetida as a prebiotic, but a brief paragraph on the known limitations of in vitro simulators versus animal or human studies would provide balance. For example:
The absence of host epithelial or immune interactions, or Potential differences in residence time and local pH gradients in vivo. Suggest framing this as an opportunity for follow-up animal or human intervention trials.
5. Consistency and Clarity in Figures and Tables
In Figure 4, please verify that “CK,” “PFP,” “WO-1,” “PFF,” and “WO-2” are defined in each panel legend for readers who may glance at individual plots.
Consider adding error bars to the bar graphs in Figures 2 and 3 to illustrate variability across replicates, especially since taxon counts can be highly variable.
In Table 1, ensure that “ND” (not detected) is spelled out in a footnote, and confirm that letter-based statistical groupings are clearly explained.
6. Minor Editorial Points- A few sentences in the Introduction and Methods could be streamlined—e.g., the lengthy listing of media components can be moved to a Supplement or tabulated. Please proofread for occasional typographical errors (e.g., “obesity-associated microbiotas” versus “obesity-associated microbiota” for subject-verb agreement).
In summary, your manuscript is methodologically sound, data-rich, and presents novel insights into the prebiotic potential of P. foetida polysaccharides and flavonoids. With these clarifications and minor additions, I believe the paper will be even more compelling and immediately useful to researchers exploring dietary modulation of the gut microbiome in metabolic disease. Congratulations on completing this important work.
Author Response
Comments 1: Clarify Donor Selection and Pooling- Please provide a brief description of donor demographics (age range, gender distribution) and any exclusion criteria (recent antibiotic use, chronic illness). In the Methods, you describe pooling fecal material from five normal-weight and five obese individuals. Adding a sentence on how you ensured homogeneity in the pool (e.g., same weight per donor, aseptic mixing protocol) will help readers reproduce the inocula preparation.
Response 1: Thank you for this helpful comment. We have added a reference that details the donor selection and description, as well as the exclusion criteria. As now indicated in revised manuscript, the weight of samples per donor was equivalent (total 40 g). We have now included that they were homogenized using a Stomacher blender.
Comments 2: Define Abbreviations at First Use- Although common in this field, please ensure that every abbreviation (e.g., PFP, PFF, SCFAs, BFBL) is defined upon its first appearance in the text.
Response 2: Thank you for this helpful comment. We have now carefully reviewed the manuscript and have defined every abbreviation (e.g., PFP, PFF, SCFAs, BFBL) upon its first use in the text, both in the main body and in the figure legends.
Comments 3: Enhance Description of Chemical Characterization- You report that PFP is enriched in galacturonic acid, galactose, arabinose, and so on, and that PFF contains 39 tentatively identified compounds. It would benefit the reader to: Include a one- to two-sentence rationale for why macroporous resin and polyamide chromatography were chosen for PFF versus PFP purification. Highlight in the main text (not only the Supplement) the three or four major flavonoid glycosides that likely drive the prebiotic effects, including their approximate relative abundance.
Response 3: We thanked the reviewer for this suggestion. We added a brief rationale for the chosen purification methods and highlighted the key flavonoid glycosides and their relative abundance in the main text as recommended.
Comments 4: Discuss Model Limitations and In Vivo Translation- Your Discussion touches on the promise of P. foetida as a prebiotic, but a brief paragraph on the known limitations of in vitro simulators versus animal or human studies would provide balance. For example:
The absence of host epithelial or immune interactions, or Potential differences in residence time and local pH gradients in vivo. Suggest framing this as an opportunity for follow-up animal or human intervention trials.
Response 4: We are grateful to the reviewer for raising this important point. In response, we have incorporated a new paragraph in the Conclusion section that acknowledges the inherent limitations of in vitro simulators compared to in vivo studies, specifically noting the absence of immune/epithelial crosstalk and physiological variables. Finally, we looked forward to the necessity of subsequent intervention trials.
Comments 5: Consistency and Clarity in Figures and Tables
In Figure 4, please verify that “CK,” “PFP,” “WO-1,” “PFF,” and “WO-2” are defined in each panel legend for readers who may glance at individual plots.
Consider adding error bars to the bar graphs in Figures 2 and 3 to illustrate variability across replicates, especially since taxon counts can be highly variable.
In Table 1, ensure that “ND” (not detected) is spelled out in a footnote, and confirm that letter-based statistical groupings are clearly explained.
Response5: We are grateful to the reviewer for these valuable suggestions to improve the clarity of our figures and table. In response:The legends for every panel in Figure 4 now explicitly define the abbreviations (CK, PFP, etc.)ï¼›Error bars have been added to Figures 2 and 3 to represent variability across replicatesï¼›A footnote in Table 1 now defines "ND" as "Not Detected," and the explanation of the statistical groupings has been clarified.
Comments 6: Minor Editorial Points- A few sentences in the Introduction and Methods could be streamlined—e.g., the lengthy listing of media components can be moved to a Supplement or tabulated. Please proofread for occasional typographical errors (e.g., “obesity-associated microbiotas” versus “obesity-associated microbiota” for subject-verb agreement).
Response 6: We thank the reviewer for these helpful editorial suggestions. We have streamlined the indicated sentences, moved the lengthy list of media components to the Supplement, and thoroughly proofread the manuscript to correct typographical errors.
Round 2
Reviewer 2 Report
Comments and Suggestions for Authors
Authors made some upgrade of the paper however based on my previous comments and suggestion they are far not enough. I can understand the answers for comments one however the precision in scientific paper should be at the first place. One-tenth is not precise. Additionally Author wrote in the response that “As now more clearly stated in the revised manuscript, this exact methodology is described in detail in References 8 and 9, which are our prior studies focused specifically on the optimization of extraction, separation, and purification processes for these exact aqueous components.”. That’s not true the procedure for methanol extraction of the material described in ref 8 is completely different. On the other hand in Ref 9. The procedure is similar however no optimisation was done there. Still we have 65oC while the time in ref. 9 is longer (4h). I believe this is a Authors mistake only not a deliberate attempt to deceive the reader.
Based on comment 2 the confirmation that a substance is polymer based on dialysis is risky. I cannot understand this especially when Authors in the revision version of the paper add the ref 9 in which some very basic molecular studies were present. Of course there is still the question of time. If the time is shorter (as in reviewed paper) the molecular weights should be lower. So if Authors have the ability to perform the experiments on molecular structure and monosaccharides content in polysaccharide backbone why it wasn’t done here? It must be done if the process parameters (extraction) is different than in ref 9. I also wonder how scientist in the scientific discussion (review process) can write that “we speculate that the remaining approximately 44% is mainly composed of bound phenolic compounds, pigments, and minerals.” There is no place for speculation here. It is time to analyse that.
Summing all this up I designate the paper still as needed major revision before publication.
Author Response
Dear reviewer,
Thank you for reviewing our manuscript again and providing these crucial and highly constructive suggestions. Your meticulous review and profound insights have greatly helped us improve the quality and rigor of the paper. We sincerely apologize for the deficiencies in the initial draft response and revision draft. We have thoroughly revised the manuscript based on each of your comments, and the detailed responses are as follows.
Comments 1: Authors made some upgrade of the paper however based on my previous comments and suggestion they are far not enough. I can understand the answers for comments one however the precision in scientific paper should be at the first place. One-tenth is not precise.
Response 1: Thank you for this helpful comment. We apologize for using such an imprecise expression as "one-tenth", as this does not conform to the norms of scientific writing. We have removed this expression from the manuscript and we have clearly specified the specific amounts of the extraction objects and extraction solvents used in the extraction process.
Comments 2: Additionally Author wrote in the response that “As now more clearly stated in the revised manuscript, this exact methodology is described in detail in References 8 and 9, which are our prior studies focused specifically on the optimization of extraction, separation, and purification processes for these exact aqueous components.”. That’s not true the procedure for methanol extraction of the material described in ref 8 is completely different. On the other hand in Ref 9. The procedure is similar however no optimisation was done there. Still we have 65oC while the time in ref. 9 is longer (4h). I believe this is a Authors mistake only not a deliberate attempt to deceive the reader.
Response 2: We sincerely apologize for the significant mistake we made in our first round of response. Due to the excessively high self-citation rate, in accordance with the editor's requirements, we mistakenly deleted reference 8, which provided a detailed description of the process of preparing AEP and PFF in our current research. At the same time, you have accurately pointed out the details of Ref 9. This method is indeed similar but with different parameters. Now we have clearly stated in the manuscript: The water extraction process used in this study is based on the method in Reference 9 but has been modified.
We also admit that due to the changes in parameters, the relevant properties of PFP obtained in this study need to be independently verified and discussed, and cannot simply be cited from previous studies.
Once again, we apologize for the inconvenience caused by this mistake. This was truly an unintentional oversight and we have taken immediate corrective action.
Comments 3: Based on comment 2 the confirmation that a substance is polymer based on dialysis is risky. I cannot understand this especially when Authors in the revision version of the paper add the ref 9 in which some very basic molecular studies were present. Of course there is still the question of time. If the time is shorter (as in reviewed paper) the molecular weights should be lower. So if Authors have the ability to perform the experiments on molecular structure and monosaccharides content in polysaccharide backbone why it wasn’t done here? It must be done if the process parameters (extraction) is different than in ref 9.
Response 3: Thank you for raising this crucial point. We agree with your view. It is insufficient to infer the properties of polymers merely through dialysis. Especially after the parameters of extraction are changed, the molecular characteristics are likely to have changed. However, we mistakenly applied the PFP research results obtained from the previous studies to the current research subjects. Therefore, we conducted an experiment for determining the relative molecular weight of PFP and included the results in the manuscript. Together with the original analysis results of the monosaccharide composition, these data were used to characterize the basic molecular structure of the substance. Meanwhile, in the discussion section, the necessity of characterizing its fine structure in subsequent research was also elaborated.
Comments 4: I also wonder how scientist in the scientific discussion (review process) can write that “we speculate that the remaining approximately 44% is mainly composed of bound phenolic compounds, pigments, and minerals.” There is no place for speculation here. It is time to analyse that.
Response 4: You have once again raised a very reasonable criticism. The use of the word "speculate" in scientific papers does indeed seem insufficiently forceful and implies a lack of evidence. We have completely removed the sentence "we speculate that..." from the manuscript.
Meanwhile, in accordance with your request, we have further analyzed the basic chemical composition of PFP, which mainly includes ash, glucuronic acid, phenolic substances, and flavonoid substances. And the detection results will be synchronously updated in the manuscript and supplementary materials.
Once again, we sincerely thank you for taking the time and effort to help us improve this paper. We hope that the above responses and the revisions made will meet your requirements. Your opinions have made our work better and more solid.